# Continuously Volumetric Rendering with Neural Density-Distance Fields

## Abstract

This paper proposes a continuous volumetric rendering and a bisection sampling utilizing the Neural Density-Distance Field (NeDDF) that can synthesize novel views with bouncing transparency during each rendering segment. Since, unlike the density field, the distance field retains the state of the nearby free space, efficient sampling, such as sphere tracing, has been attempted by assuming a solid object. However, distance fields struggle to represent transparency, detailed shapes, and distant landscapes. We derive bounds on transparency in the interval in volume rendering based on NeDDF, which extends distance fields to non-solids. Through realizing the derivation, we invent an efficient bisectional exploratory sampling method that minimizes the maximum of the bound range. For scaling to fit the Eikonal constraints on distance fields, Multi-resolution Hash Encoding, which is excellent for detailed description, is used with frequency separation. We achieve unmasked acquisition of scenes with distant scenery by introducing contract coordinates and scaling the distance field so finite values can describe it. Experiments on synthetic and real data show that the proposed rendering bounds work reasonably.

## 1 Introduction

Combining a shape representation called Neural Fields (NF) (Xie et al., 2021) and differentiable volume rendering leads to breakthroughs in 3D restoration from multiple viewpoint images. NeRF (Mildenhall et al., 2020), a leading example, is a NF that regresses the density field, which takes the 3D position as input and outputs density, and the color field. NF can acquire dense field representations, including detailed shapes and direction-dependent colors, in much smaller models than conventional voxel-based representations.

The distance field, another way to describe the geometry, expresses object shape by the distance to the closest object surface from the input position. While the density field has its gradient only inside the object, the distance field holds the distance to objects even outside them, and it has implicitly holds direction as its gradient. Therefore, voxel-based distance fields like TSDF (Curless & Levoy, 1996) have been used for shape integration(Newcombe et al., 2011) and non-rigid scene-tracking (Newcombe et al., 2015; Li et al., 2020) applications. Due to its potential utility for these tasks, applying NF to distance fields has been attempted (Park et al., 2019; Chibane et al., 2020; Oechsle et al., 2021). In particular, approaches that regress distance fields on NF and provide conversion formulas from distance to density allow both distance and density fields to be learned similarly to NeRF. For example, NeuS (Wang et al., 2021) assumes that the density is normally distributed around the zero level of the signed distance field and then calculates the density from the distance and variance parameters. NeDDF (Ueda et al., 2022) introduces Density-Distance, a non-solid extension of the unsigned distance field, and calculates the density from the distance and its gradient.

There are two issues regarding convergence in acquiring distance field representations with NF. (a) Due to its high spatial dependence, the shape representation using NF is difficult to regress with local conditioning. Using local feature approaches have achieved high effectiveness in this problem (Sun et al., 2022; Fridovich-Keil et al., 2022; Liu et al., 2020; Müller et al., 2022) for density fields, increasing the resolution of the grid requires the differentiable compression of parameters such as octree (Fridovich-Keil et al., 2022), tensor decomposition (Chen et al., 2022), and hash

tables (Müller et al., 2022). However, grid-based parameterization cannot apply naively to distance fields because grid-based methods allow large amplitude for high-frequency components and they require sparseness. Distance field is limited the gradient strength (not high-frequency) and not sparse in general. Another issue is that (b) the signed distance fields do not allow for the representation of thin shapes, such as leaves or wires, because they assume watertight shapes. NeDDF eases this problem by basing it on the unsigned distance field, but it tends to blur color textures. Since it is difficult to smooth the density field in the distance field, general coarse-to-fine sampling is unstable for sparse sampling to find object regions, leading to insufficient samples around object contours.

We propose solutions to these problems based on NeDDF. (a) To acquire the high-frequency components of the distance field, we take the approach of separating the features by frequency bands and adapting the scale of the high-frequency components to the distance field representation. The proposed method obtains features separated by frequency bands using Multi-resolution Hash Encoding. Restricting the amplitude of the features in the high-frequency band enables the learning of grids compatible with the constraints of the distance gradient. Since the amplitude is more restricted for higher resolution grids, the features of the higher resolution grids have less influence on the density of NeDDF in regions where the distance outside the object is significant, and the effect of hash table collision is unlikely to occur. Therefore, our approach can apply the grid-based architecture to NeDDF, which is distance-field based. (b) To recover thin objects, we take the approach of using the properties of NeDDF to infer the conditions between two points on a ray. We derive the maximum and minimum light transmittance values at a given interval on the ray, considering that the maximum value of the gradient amplitude is 1 in NeDDF. The proposed method places the sampling points recursively using the bound of the influence on the rendering color. With an assumption on the gradient of the distance field at each sampling point, we tightly bound the transmittance of each interval from above and below, even with a small number of sampling points.

Our contributions are as follows. (i) We derive a lower bound on the transmittance that can guarantee an upper bound on the color weights of the interval from sampling on distance field information. We also provide a sampling method that optimally divides the color weights. (ii) Using an assumption on the gradient of the distance field, we bound the transmittance from above and below to achieve volume rendering that does not require discretization of the density. (iii) We propose a fast architecture that adapts to both color and distance fields with different frequency characteristics by acquiring feature frequencies separately using Multi-resolution Hash Encoding and a small MLP with masks.

## 2 RELATED WORK

In recent years, the methods called Neural Fields (NF) (Xie et al., 2021) which directly represent continuous signals has been attracting significant interest in the research community. Given sufficient parameters, fully connected neural networks can describe continuous signals over arbitrary dimensions, as the representative method Neural Radiance Fields (NeRF) (Mildenhall et al., 2020).

### 2.1 NEURAL DISTANCE FIELD

A distance field is a shape representation that takes a 3D position as input and outputs the distance to the nearest neighbor boundary. In contrast to density fields, they are helpful for shape integration and camera pose estimation tasks, as they retain information about the direction and distance of the object, even outside the object. In addition, as it is based on level set functions, boundary surfaces can be easily extracted using like the Marching Cubes method. On the other hand, the distance field assumes a boundary surface, which restricts the subject to solid objects. In addition, distance fields need to satisfy complex constraints (called Eikonal) but it is difficult to integrate to training (Sharp & Jacobson, 2022).

A model that addresses these challenges while maintaining the usefulness of distance-field-based methods is the NeDDF, which extends the Unsigned Distance Field to semi-transparent scenes by linking distances and their gradients to densities. Specifically, significant minima of Density-Distance and gradients smaller than one are interpreted as small densities. This makes it possible to use the system even for scenes containing semi-transparent or spatially high-frequency objects and solid scenes, as the learning process that does not satisfy the Eikonal constraint can be interpreted as a blurred density distribution, which provides stability training. On the other hand, NeDDF has a

problem with applying methods such as the sphere tracing, which focuses on sampling points near the boundary surface, to low-density objects. In this study, we clarify that transmittance bounds can be established from sampled points in NeDDF without discretizing the density.

## 2.2 GRID-BASED NEURAL FIELDS

Many NF, including NeRF, have used a coordinate-based MLP that combines Positional Encoding and MLP to incorporate high-frequency components. Despite using Positional Encoding to capture high-frequency components, it is slow to acquire local features and takes a long time to learn a scene. Hybrid approaches that combine MLP with a 3D grid representation so that local features can be conditioned are being worked on for faster learning (Hedman et al., 2021; Liu et al., 2020; Sun et al., 2022).

Grid-based methods are limited in resolution because the number of parameters requires a memory capacity of $O(N^3)$ for resolution of each dimension $N$. For models that deal with density fields, approaches that use differenciable compression methods, such as octree-based methods (Fridovich-Keil et al., 2022), tensor decomposition methods (Chen et al., 2022), and wavelet transform methods (Rho et al., 2022), have been proposed to handle high resolution with small memory footprints. However, these compression methods take advantage of the sparseness of density fields, and it is difficult to compress information in the distance field, which retains information even outside the object. We focus on Multi-resolution Hash Encoding (Müller et al., 2022) as a grid-based representation that can be expected to provide sufficient compression even for distance fields. It uses concatenated $L$ features that spatially interpolated features from each resolutions $[N_1, ..., N_L]$. In this method, the reference for each grid is determined by a Hash Table of table size $T$. When describing a density field, the low-resolution grid can hold values densely because the number of elements is less than $T$, and the high-resolution grid is sparse except at object boundaries, so Hash collisions are unlikely to occur. When using distance fields, the amplitude of high-frequency components is suppressed due to the limitation of the magnitude of the gradient of the distance field. In particular, in NeDDF, the effect of high-frequency components with small amplitudes is reduced in the areas where the distance outside of the object is large. In other words, Multi-resolution Hash Encoding is effective for NeDDF compression because sparsity is practically valid for high-resolution grids if the grids with small resolution can adequately represent low-frequency components.

## 3 METHODOLOGY

### 3.1 PRELIMINARY: NEURAL DENSITY DISTANCE FIELD

This method is based on NeDDF to improve quality and latency of learning and inference. Thereafter, we consider the ray $\mathbf{r}(t) = \mathbf{p} + t\mathbf{v}$ from point $\mathbf{p}$ into the direction $\mathbf{v}$. NeDDF introduces the following Density-Distance $D(\mathbf{p})$, which is a non-solid extension of the unsigned distance field:

$$D(\mathbf{p}) := \min_{\mathbf{v} \in S^2} \left( t_n + \int_{t_n}^{t_f} tT(t)\sigma(\mathbf{r}(t)) dt \right).$$ (1)

From the density distance and its gradient, the density information is obtained by follow:

$$\sigma(\mathbf{p}) = \frac{1 - \|\nabla D(\mathbf{p})\|_2}{D(\mathbf{p})}.$$ (2)

In the volume rendering, NeRF (Mildenhall et al., 2020) aggregates color information as following equation.

$$\hat{\mathbf{c}} = \int_{t_0}^{t_n} T(t)\sigma(t)\mathbf{c}(t) dt, T(t) = \exp\left( -\int_{t_0}^{t} \sigma(s) ds \right).$$ (3)

In NeDDF, like other density-based methods, volume rendering assumes that the density is constant in each interval and aggregates colors as following discretization (Mildenhall et al., 2020):

$$\hat{\mathbf{c}} \simeq \sum_{i=0}^{n} T_i (1 - \exp(-\sigma(r_i)(t_{i+1} - t_i))) \mathbf{c}(r_i, v).$$ (4)

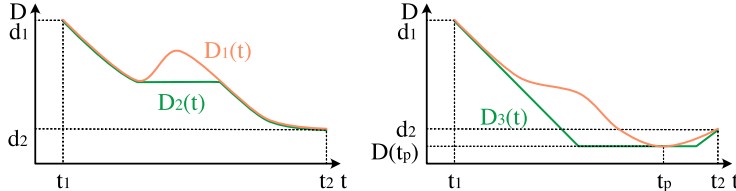

Figure 1: (Left) Trajectory $D_1$ with an interval with strictly monotonically increasing and strictly monotonically increasing transmittance has a trajectory $D_2$ with a smaller transmittance. (Right) Minimum transmittance trajectory $D_3$

Let $t = t_0, \ldots, t_n$ on ray $\mathbf{r}(t)$ with $n+1$ sampling points $r_0, \ldots, r_n$ on $t = t_0, \ldots, t_n$. Hereafter, to deal with the model on rays, we define the distance field for the scalar $t$ by $D(t) = D(\mathbf{r}(t))$. Also, we let a component of $\nabla D(\mathbf{r}(t))$ in the $\mathbf{v}$ direction as $g(t) = \frac{\partial D(\mathbf{r}(t))}{\partial t}$ and a vertical component as $h(t)$ which satisfies $|\nabla D(\mathbf{r}(t))|^2 = \sqrt{h(t)^2 - g(t)^2}$.

## 3.2 Continuous Volumetric Rendering

This section describes a rendering method that uses distance field information. In volumetric rendering, the transmittance $T(t)$ is essential in determining the color blending factors. However, since the true $T(t)$ is unavailable, previous methods discretize it with the inappropriate assumption that the density is constant between sampling points. We consider bounding $T(t)$ from above and below based on the sampled information, utilizing the property that the trajectory of the distance field can be limited.

**Transmittance Bound**   Constraints on the norm of the distance gradient can limit the trajectory between two sampled points. We derive the trajectory between the two sampled points that minimizes the transmittance. As shown in Fig.1, we consider the case that two sampled points at $t = t_1$ and $t = t_2$ on the ray give distances $d_1$ and $d_2$. The transmission of light $T_1$ in the interval $[t_1, t_2]$ is calculated as follows.

$$T_1 = \exp\left(-\int_{t_1}^{t_2} \frac{1 - \sqrt{h(t)^2 + g(t)^2}}{D(t)} dt\right) \tag{5}$$

Let $D_{min}$ be the trajectory of $D$ in which the transmittance of $T_1$ in the interval is minimal. We can consider $h(t) = 0$ in $D_{min}$, since the density is maximum when the gradient of the distance field has no ray and vertical components. We first show that $D_{min}$ follows a monotonically decreasing then monotonically increasing trajectory in the interval. In trajectories that do not satisfy the above, such as $D_1$ in Fig.1(left), there are intervals with strictly monotonically increasing and strictly monotonically increasing transmittance. There are trajectories such as $D_2$ that horizontally connect such intervals, which always holds $D_1(t) \geq D_2(t), \frac{\partial D_1(t)}{\partial t} \geq \frac{\partial D_2(t)}{\partial t}$. From the equation 2, there are trajectories $D_2$ with smaller transmittance in $D_1$, as the density is always below $D_2$ in $D_1$. Therefore, as shown in Fig.1(right), $D_{min}$ is broadly minor decreasing at $t < t_p$ and broadly minor increasing at $t_p < t$, for $t_p \in \arg\min_t D(t)$. The $T_1$ is the following calculation:

$$T_1 = \exp\left(-\int_{t_1}^{t_p} \frac{1 + g(t)}{D(t)} dt - \int_{t_p}^{t_2} \frac{1 - g(t)}{D(t)} dt\right) = \exp\left(-\int_{t_1}^{t_2} \frac{1}{D(t)} dt\right) + \frac{d_1 d_2}{D(t_p)^2}. \tag{6}$$

Since $T_1$ becomes smaller as $D(t)$ becomes smaller, $D_{min}(t)$ always takes the smallest value within the constraint. Therefore, $D_{min}$ consists of a straight line with slope -1,1 from both ends and a straight line with $D(t) = D(t_p)$, like $D_3$ in Fig.1. The minimum transmittance is calculated as follows:

$$T_1 = \exp\left(-\int_{t1+d_1-D(t_p)}^{t_2-d_2+D(t_p)} \frac{1}{D(t_p)} dt\right) = \exp\left(-\frac{t_2 - t_1 - d_2 - d_1}{D(t_p)} - 2\right) \tag{7}$$

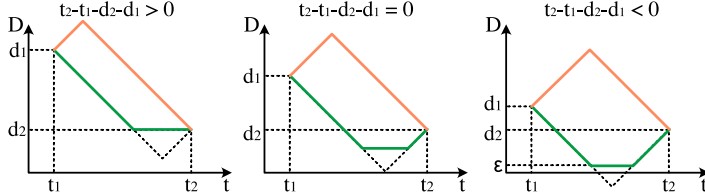

Figure 2: The trajectory of maximum (orange) and minimum (green) transmittance through the two points.

As shown in Fig.2, $D(t_p)$ in which $T_1$ is minimized is divided into the following three cases depending on the sign of $T_2 - T_1 - D_2 - D_1$, i.e., the trajectory $D(t)$ can take touching the $t$ axis.

$$\begin{cases} D(t_p) = \min(d_1, d_2) & (t_2 - t_1 - d_2 - d_1 > 0) \\ D(t_p) \in [\epsilon, \min(d_1, d_2)] & (t_2 - t_1 - d_2 - d_1 = 0) \\ D(t_p) = \epsilon & (t_2 - t_1 - d_2 - d_1 < 0) \end{cases} \quad (8)$$

Note that $\epsilon$ is a very small scalar representing the minimum distance field.

**Sampling** This section describes a bisection approach that recursively divides the sampling interval using lower bounds of transmittance. Our important insight is that volume rendering can efficiently select sample points by introducing a divide-and-conquer manner.

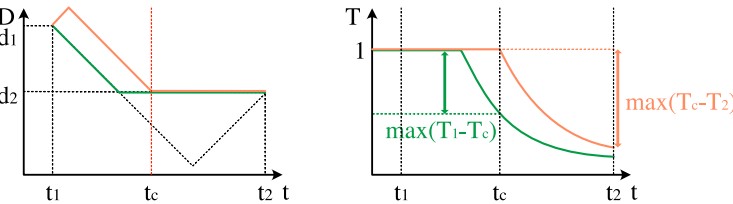

Figure 3: Distance fields (left) and transmittance (right) that maximize the respective transmittance when the interval is splitted by $T_c$. The green and orange trajectories maximize $T_1 - T_c, T_c - T_2$.

We consider the partitioning of the sampling interval $t_1, t_2$ at $t_c$. Since there is no limitation on the color field, aggregating colors for volume rendering requires the assumption that the color is constant in sampling intervals. Sampling $t = t_1, t_c, t_2$ yields the colors $\mathbf{c}_1, \mathbf{c}_c, \mathbf{c}_2$. From equation 3, the rendered color $\hat{\mathbf{c}}$ is as follows:

$$\hat{\mathbf{c}} = (T(t_c) - T(t_1))\mathbf{c}_1 + (T(t_2) - T(t_c))\mathbf{c}_c. \quad (9)$$

In practice, errors occur since colors are not constant in the interval. Reducing the color coefficients $T(t_c) - T(t_1), T(t_2) - T(t_c)$ leads to smaller errors. Since the coefficients cannot be minimized directly in $t_c$, we minimize respective upper bounds. To decrease both coefficients, $t_c$ lies in the flat range of Fig.2. Therefore, the trajectory of the distance field and the transmittance that maximizes each coefficient are determined as shown in Fig.3. The green trajectory with the minimum transmittance maximizes $T(t_c) - T(t_1)$, as shown in Fig.2. The orange trajectory that makes $T(t_c) - T(t_1)$ take 0 maximizes $T(t_2) - T(t_c)$.

The optimal $t_c$ gives $\max(T(t_c) - T(t_1)) = \max(T(t_2) - T(t_c))$. From Fig.2, we obtain $t_c = \frac{1}{2}(t_1 + t_2 + d_1 - d_2)$ for all conditions. Therefore, our sampling method recursively splits the sampling interval in such a way as shown in Fig.4. Note that we use a tighter bound with assumptions to determine the end of sampling, which will be discussed later.

**Tighter Bound of Transmittance with Assumption** For rendering and determining the end of sampling, we derive a tighter bound of the interval transmittance. The maximum transmittance in the interval is 1 on trajectories that always have a gradient of 1, such as in Fig.2, which is inconvenient

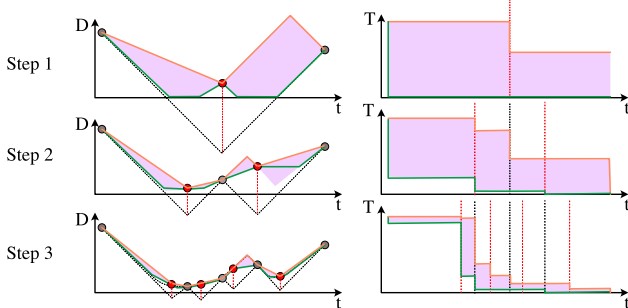

Figure 4: Sampling Method. (left) We divide intervals at each sampling step using the intersection of two ends of the interval extended by the gradient -1,1 for the distance field as indicated by the red dots. The obtained sampling points restrict paths to the purple area. The path with the green line takes the minimum transmittance value, and the orange line takes the maximum. (Right) Upper bound (orange) and lower bound (green) of the transmittance for each interval.

for determining the end of sampling. Assuming that the second derivative of the distance field $\frac{\partial g(t)}{\partial t}, \frac{\partial h(t)}{\partial t}$ does not cross zero in the interval allows the useful upper bound of transmittance. This assumption holds well when both sides of the interval refer to the same object, and fits better as the number of sampling points increases.

Due to equation 2, only the density depends on $h(t)$, and the density is monotonically increasing concerning $h(t)$. To calculate the maximum and minimum transmittance values, we can fix $h(t)$ at the minimum and maximum values of $h(t_1), h(t_2)$, respectively. The trajectory of the distance field in the ray direction is bounded by the trajectory $D_\alpha$ extended at both ends by the gradients $g(t_1)$ and $g(t_2)$ and the trajectory $D_\beta$ directly connected at both ends, as shown in Fig.5. The $D_\gamma$ is a similar trajectory to the minimum transmittance in Fig.2, and becomes a candidate for bounding of transmittance when it is within the above range. Transmittance maxima and minima appear in $D_\alpha, D_\beta, D_\gamma$. We determine the end of sampling by whether supremum of the difference between the maximum and minimum transmittance values obtained from these exceed a threshold value. When rendering, we assume the bounding range is sufficiently tight and use the trajectory with the maximum transmittance.

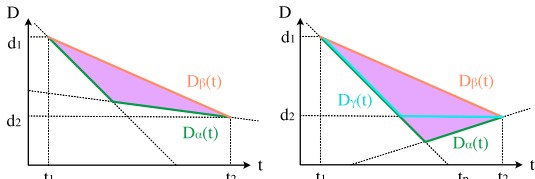

Figure 5: The assumption of a range of gradients restricts the distance field to the purple area. The maximum and minimum transmittance values appear in $D_\alpha, D_\beta, D_\gamma$. (left)$g_1 g_2 > 0$. (right)$g_1 g_2 \leq 0$.

### 3.3 MULTI-RESOLUTION HASH ENCODING

In this section, we describe how to induce the acquisition of frequency band-separated features from Multi-resolution Hash EncodingMüller et al. (2022) and adapt them to the representation of distance fields. Unlike density and color fields, distance fields have the restriction that the gradient norm is less than 1. Therefore, true distance fields allow smaller amplitudes for high-frequency components. To fit features to distance and color fields requires separating them by frequency bands. We focus on the fact that features obtained from Multi-resolution Hash Encoding have different Nyquist frequencies for each resolution. A small MLP with masked weight matrices produces features obtained from grids that are separated by frequency bands such that only features derived from lower-frequency grids are captured. The specific architecture is shown in Fig.6. Multi-resolution Hash Encoding

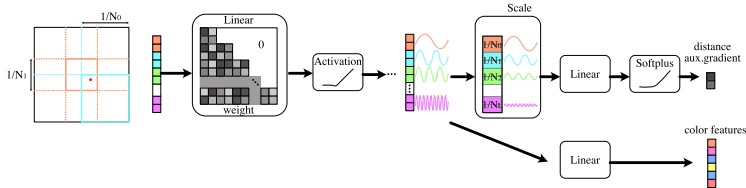

Figure 6: Architecture of Multi-resolution Hash Encoding and network

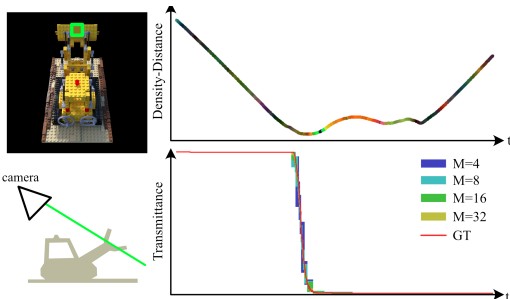

Figure 7: Example of transmittance bounds for pixels indicated by the green box. (Top) Graph of distance field values on a ray for dense sampling. The colors are the color field values for each sampling point. (Bottom) Graph of transmittance on a ray. Each colored area shows the range of transmittance bounds for each threshold value. The red line is the transmittance calculated from dense sampling.

parameterizes $F$-dimensional features in L grids with a resolution of $N_1, \cdots, N_L(N_1 < \cdots < N_L)$ on each axis. Multi-resolution Hash Encoding outputs a vector of $L \cdot F$-dimensions ordered by resolution. The linear combination layer keeps the Nyquist frequency of each feature component by masking the upper triangular elements of the weight matrix with zeros, as shown in Fig.6.

We consider the scale to match the distance field of a feature $f_i$ output by a grid $i$ with a resolution of $N_i$ per axis. Note that although we treat it in one dimension for the sake of explanation below, in practice, interpolation is performed by multiplying the coefficients of each axis in three dimensions. During inference of the grid, smoothstep function(Müller et al., 2022) interpolates the first-order derivatives of the features to make them continuous with input positions. The smoothstep function gives the mixing factor of the features at the grid vertices by $S_1(x) = x^2(3 - 2x)$ using the fraction $x \in [0, 1]$ within grid i containing the input point $p$. This coefficient gives the interpolated feature $f_i$ and gradient $\frac{df_i}{dx}$. Since the size of one grid in the original coordinate space is $1/N_i$, thus $\frac{dx}{dp} = N_i, \frac{df_i}{dp} = N_i \frac{df_i}{dx}$. Suppose the value ranges of the parameters in each grid are close. In that case, the norm of the feature gradient is proportionate to $N_i$ since the smoothstep function is independent of the grid resolution. For example, the features output by grids with resolutions of 16 and 512 will have x32 scale differences. In the distance field description, both features should have high expressivity in the range of gradients below 1, so we prefer a scale that multiplies $f_i$ to $1/N_i$. On the other hand, the color field has no restriction on high-frequency components. Thus, it is ideal to treat it on an equal scale. A single network can stably handle the distance and color fields by applying such a scale to the output of the MLP, as shown in Fig.6.

This frequency separation also allows the use of distance fields in unbounded (360 degrees around a point) scenes. For example, when using the contract coordinate system, we scale distance field value $d$ into $d_{scaled}$ to fit the scale of the gradient as follows for the input position $\mathbf{p}$:

$$d_{scaled} = \begin{cases} d & (\|\mathbf{p}\| <= 1) \\ \|\mathbf{p}\|^2 d & (\|\mathbf{p}\| > 1) \end{cases} \tag{10}$$

Table 1: Quantitative evaluation on synthetic dataset. We report PSNR (higher is better).

| Method | Chair | Drums | Ficus | Hotdog | Lego | Materials | Mic | Ship | Mean |
|---|---|---|---|---|---|---|---|---|---|
| NeRF(hours) | 33.00 | 25.01 | 30.13 | 36.18 | 32.54 | 29.62 | 32.91 | 28.65 | 31.01 |
| iNGP(5 min) | 35.00 | 26.02 | 33.51 | 37.40 | 36.39 | 29.78 | 36.22 | 31.10 | 33.18 |
| NeuS(hours) | 27.69 | 22.14 | 21.67 | 32.14 | 27.18 | 25.64 | 27.52 | 23.47 | 25.93 |
| NeDDF(hours) | 29.11 | 23.96 | 25.72 | 30.85 | 27.93 | 25.52 | 29.34 | 23.69 | 27.02 |
| Ours(10 min) | 30.14 | 25.78 | 27.33 | 33.24 | 29.75 | 28.86 | 31.57 | 26.55 | 29.15 |

Table 2: Quantitative evaluation on Mip-NeRF 360 dataset.

| Method | PSNR($\uparrow$) | SSIM($\uparrow$) | LIPIS($\downarrow$) |
|---|---|---|---|
| Mip-NeRF 360+iNGP | 25.58 | 0.804 | 0.160 |
| Zip-NeRF | 32.52 | 0.954 | 0.037 |
| Ours | 26.72 | 0.861 | 0.103 |

## 4 EXPERIMENTS

### 4.1 EVALUATION OF IMAGE QUALITY

**Experimental setup.** We quantitatively evaluate the video generation quality of our method using the NeRF synthetic, mip-nerf360 dataset. We compare NeRF (Mildenhall et al., 2020), iNGP(Müller et al., 2022), NeuS(Wang et al., 2021), and NeDDF(Ueda et al., 2022) as baselines for synthetic data. For the 360 data set(Barron et al., 2022), we use the grid-based Mip-NeRF 360+iNGP and Zip-NeRF as baselines. We refer to the values in the Zip-NeRF paper(Barron et al., 2023). NeuS, NeuS2, and NeDDF are omitted because they did not converge for the unmasked 360 scenes.

**Result.** Table 1 shows the PSNR for each scene and each method for evaluating the quality of the generated images. Table 1 shows the image generation quality of each method. Our method improves the image quality for all scenes from the conventional distance-field-based methods, NeuS and NeDDF. Fig.8 shows a comparison of the rendering results in areas that are difficult to recover with the distance field-based methods. It is challenging for distance-field-based methods to describe complex topological shapes with fine holes, such as in Lego and Mic. Therefore, in NeuS, the shape is smoothed to fill the holes, and in NeDDF, the density near the contour is smoothed. In contrast, our method improves the quality of the generated image in the area of the fine holes. We consider that using grids has enabled local conditioning of the details, which is effective. In addition, NeDDF had difficulty capturing fine color changes in scenes with strong specular reflection, such as the curved face of Mic and the water in Ship. NeDDF reduces high-frequency components in the entire network to suit the distance field. Our method achieves detailed color representation by frequency separation of features and varying the intensity of each frequency band between the distance and color fields. Moreover, distance field-based conventional methods have difficulty acquiring thin shapes such as legs of drums, leaves of ficus, and wires of ships. Our proposed method greatly improves the quality of such details. We suppose that the proposed sampling and rendering interpolated from the distance field improved the quality of the image because it provided sufficient resolution for optimizing the distance field. Table 2 also shows the video quality in 360 real scenes. The proposed method can provide effective gradients even when the object exterior is wide due to the improved sampling method. The proposed method performs comparably to iNGP even in 360 scenes without foreground masks, which is difficult to obtain with conventional distance-field-based methods.

### 4.2 EVALUATION OF SAMPLING

**Experimental setup.** We confirm that the proposed sampling can bound the transmittance and confirm the convergence of the bounds. The experiment uses 1024 rays randomly selected from a test camera viewpoint for a model already trained on the Lego scene. For the ground truth value of transmittance, we use the value computed by volume rendering from a sufficiently dense 2048 equally spaced sampling, assuming constant density at each interval. For a resolution threshold of

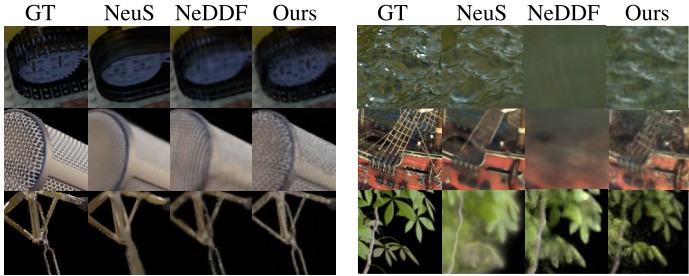

Figure 8: Rendering results of each method in areas that are challenging to restore with distance-field

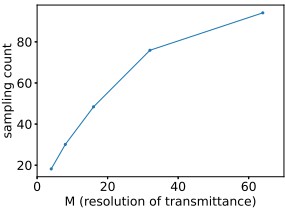

Figure 9: The plot of the number of samplings satisfying the resolution of transmittance $M$.

$M \in [2, 4, 8, 16, 32]$ of transmittance, divide until the difference between UpperBound and LowerBound is $1/M$. We verify that the UpperBound and LowerBound of the transparency are between the true values. Also, measure the average number of samplings where the range of bounds is below the threshold for each M.

**Result.** The true transmittance values ranged within the bounds for all the lays used in the verification. An example of a ray is visualized in Fig.7 It can be seen that even for high densities, i.e., rapid transmittance changes, as in a solid scene, a segmentation is achieved such that the bounds are below the threshold. In addition, the distance field in the sampling interval is in the form of multiple increases and decreases, and the assumption that the two derivatives of the distance do not cross zero is not satisfied in the initial segmentation, but the segmentation succeeds in correctly bouncing the transmittance because it is divided into intervals that satisfy the assumption faster than the convergence of the bouncing range. The Fig.9 also shows the relationship between the resolution $M$ of the transmittance and the number of samplings when the section is divided until it satisfies the assumption.

## 5 DISCUSSION

The proposed method has excellent performance in generating new viewpoints among the methods using distance fields. On the other hand, compared to models using density fields, the proposed method is slightly inferior to iNGP, which uses the same Multi-resolution Hash Encoding, in terms of accuracy, and convergence takes about twice as long. This is due to the limitation of the distance field, which provides the distance and direction to nearby objects outside the object, compared to the density field, which provides only the presence or absence of objects. Unlike density fields, where it is sufficient to place sampling points only at object boundaries, learning distance fields requires sampling points outside the object as well. The consideration of the gradient of the distance field with respect to its 3D location not only increases the complexity of the parameter space, but also increases the inference time of the network. In implementation, the need to provide continuous and smooth first-order derivatives makes it difficult to use fast activation functions such as ReLU or low-precision parameters such as FP16.

This paper introduces a lower bound on transmittance for sampling and a tighter bound that requires assumptions for rendering and determining the end of sampling. Frequency separation and scaling also allow for fast and stable use of distance fields with a bound on the gradient norm.

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
