# OpenReview forum: "Continuously Volumetric Rendering with Neural Density-Distance Fields"
_ICLR.cc/2024/Conference — Submitted to ICLR 2024_

### Official Review · Reviewer_mezv · 2023-10-16

**Soundness:** 2 fair
**Presentation:** 1 poor
**Contribution:** 2 fair
**Rating:** 1
**Confidence:** 3

**Summary:**

The paper proposes an efficient sampling strategy for neural density fields to solve novel view synthesis-type problems. The key innovation is a a lower bound on the transmittance based on gradient of the distance field. Using the lower bound and a recursive sampling scheme the authors propose to divide the transmittance function into segments that enable an effective approximation of the underlying true continuous transmittance for rendering.

**Strengths:**

The paper formulated a novel quantitive criteria for effective sampling of transmission functions that are based on distance fields and their gradients. The authors show some improvement on some synthetic test cases.

**Weaknesses:**

I think this work is not yet ready for publication for the following reasons:

1. Lack of clarity and context
The writing, problem definition, methodology and conclusions all lack clarity and context.

(a) At a high-level my key issue is that the authors do not highlight and motivate the use of neural density fields for rendering. While the authors propose a solution to a subproblem of sampling it is not clear to me how relevant this problem is to rendering at large and the results do not look promising as compared to e.g. ngp density rendering.

(b) The methodology itself is not well motivated or connected to prior art. Lack of citations and explicit statements about novelty make it hard to read and interpret section 3. What is new and what was already perviously proposed? This should be clarified in the appropriate place in Section 3.

(c) The abstract is confusing and hard to interpret without reading the full paper. Furthermore it is misleading -- it proposes that the approach helps with rendering transparency or landscapes yet this is not highlighted or supported by any experiments.

(d) all of the figures need more explanations and clarity. Why did the authors choose the specific figures? What are these figures highlighting? For example, what do we learn from figure 7 top panel? All of the colors are mixed to an extent where I cannot differentiate and the caption does not explain why this figure is there to begin with. The same goes for bottom panel of figure 7, where M=32 is no longer visible and the overall impact of choosing different M on rendering is a complete mystery.

(e) The related and prior work (and perhaps even experiments) should also include a discussion of more recent plane-based neural fields like [1,2].

[1] Fridovich-Keil, Sara, et al. "K-planes: Explicit radiance fields in space, time, and appearance." Proceedings of the IEEE/CVF Conference on Computer Vision and Pattern Recognition. 2023.
[2] Shue, J. Ryan, et al. "3D neural field generation using triplane diffusion." Proceedings of the IEEE/CVF Conference on Computer Vision and Pattern Recognition. 2023.

2. Lack of validation and experimentation
The paper only shows some minor quantitative improvement in 8 synthetic tests and lack of improvement on larger datasets as compared to ngp. Furthremore:
(a) it is unclear how the synthetic examples were chosen? were they picked to show case anything? were they randomly selected etc
(b) I think a simple baseline with voxelized distance fields is needed for comparison
(c) experiments are not well described. e.g. how was the evaluation preformed? how many view angles were used and for far are they from the training view angles etc? I understand that perhaps these details are described in the original NeRF paper but this paper should be somewhat self-contained at least as far as experiment details go.

Minor details:
- The subscript 'n' (or 'f') are not defined in Eq (1) or near it. In fact 'n' is then used for Eqs (3-4) but only defined following Eq (4).
- Eq (3) derived from the rendering equation which *far* predates NeRF (cited near Eq 3). It is worth giving readers the correct historical context which dates back at least to Kajiya et al (1984) but could probably be traced further back (out of computer graphics) to radiative transfer by Chandrasekhar (1950)

**Questions:**

My main critique is summarized in the previous section. I think that the manuscript is not mature enough for publication in the current conference but perhaps my concerns could be address for future submissions.

---

### Official Review · Reviewer_vVNf · 2023-11-01

**Soundness:** 2 fair
**Presentation:** 2 fair
**Contribution:** 2 fair
**Rating:** 3
**Confidence:** 4

**Summary:**

This paper proposes a sampling technique for the Neural Density-Distance Fields (NeDDF). NeDDF tends to produce blurry textures as it's difficult to find the exact object boundary via sparse points sampling along the query ray.  The paper introduces a divide-and-conquer sampling approach based on lower bounds of transmittance to select sample points efficiently. It derives a lower bound on transmittance using distance field information and provides a sampling method that optimally divides the color weights. The paper also proposes a strategy to effectively combine multi-resolution hashing with the density distance fields. The authors validate their method on the Synthetic-NeRF and Mip-NeRF 360 datasets and show better rendering quality. Overall, the paper is an incremental work of NeDDF to improve its rendering qualities.

**Strengths:**

- The analysis of transmittance bound and divide-and-conquer sampling is interesting.
- The experiment results indeed show improvement over NeDDF.

**Weaknesses:**

-  The major concern is that the paper seems like an incremental work on NeDDF, with its main target being to improve rendering quality. There are many recent works that report better metrics than NeuS and NeDDF, covering both density and distance-based representations, such as 3D Gaussian Splatting and Voxurf. It is unclear how the proposed approach compares with these state-of-the-art methods, making it challenging to evaluate its position since the paper is very focused on NeDDF.

-  The main advantage of NeDDF is its ability to handle shapes with no explicit boundary, such as fur or smoke. It is unclear whether this advantage is retained in the proposed approach.

-  Naively, one could sacrifice time and densely sample along the ray to achieve better rendering results. Is the proposed sampling strategy always better than naïve solutions? Moreover, we could use the same number of samples as the proposed approach and evenly distribute or weighted sample the points according to distance values. How does this strategy compare with the proposed method?

- Combining multi-resolution hashing with the distance field has been well addressed by Voxurf. The novelty of Section 3.3 is vague, and both qualitative and quantitative comparisons with Voxurf are essential to demonstrate the strength.

References:
3D Gaussian Splatting: '3D Gaussian Splatting for Real-Time Radiance Field Rendering.'
Voxurf: 'Voxurf: Voxel-based Efficient and Accurate Neural Surface Reconstruction.

**Questions:**

Please see the weakness

---

### Official Review · Reviewer_81Bz · 2023-11-01

**Soundness:** 2 fair
**Presentation:** 2 fair
**Contribution:** 2 fair
**Rating:** 3
**Confidence:** 4

**Summary:**

This paper defines NeDDF, a distance-density field, to derive bounds on transparency in the interval. It can facilitate the point sampling along the ray during volume rendering.

**Strengths:**

It is nice to have a transmittance bound for each interval along the ray to facilitate the point sampling.

**Weaknesses:**

1. Experimental results does not verify the advantage of the proposed NeDDF in Volumetric rendering. As shown in Table 1&2, the PSNR and SSIM of NeDDF is not superior to STOA methods.
2. The symbols used in this paper is somewhat confusion. For instance, the vertical component h(t) is introduced but without intuitive explanation.

**Questions:**

The SDF to density conversion in NeuS is designed to obtain high-quality geometric reconstruction from multi-view images.  NeDDF is designed according to UDF in a similar way.  Why are there no geometric reconstruction results to show the advantage of the proposed method?

---

### Meta-Review · Area_Chair_ZRY8 · 2023-12-08

**Metareview:**

This paper introduces an efficient sampling technique for neural density-distance fields (NeDDF). In the proposed divide-and-conquer sampling approach, a lower bound is derived on transmittance, which is used to select sampling points efficiently. Additionally, the paper suggests a strategy to integrate multi-resolution hashing with density distance fields. Based on experiments conducted on synthetic datasets, results indicate that the proposed approach improves rendering quality.

The paper formulates a novel quantitative criterion for the effective sampling of transmission functions based on distance fields and their gradients. The analysis of the transmittance bound and the divide-and-conquer sampling approach both contribute interesting insights. Experimental results indicate that NeDDF performs better than NeDDF in some synthetic test cases.

However, the experimental validation falls short of providing sufficient and convincing evidence. While minor quantitative improvements are observed on eight synthetic tests, the selection criteria for these examples are unclear. Furthermore, the proposed method's performance, as indicated by PSNR and SSIM in Tables 1 and 2, does not surpass some compared methods. There is also a notable absence of a comparison with a basic baseline using voxelized distance fields. The key advantage of NeDDF lies in its ability to handle shapes with no explicit boundaries, such as fur or smoke; however, it is unclear whether the proposed approach retains this capability. The paper is positioned as an incremental improvement to NeDDF. However, recent methods like 3D Gaussian Splatting and Voxurf have demonstrated superior performance over NeDDF. The comparison of the proposed method with these state-of-the-art techniques is conspicuously absent. Concerns with exposition include a lack of clarity and context, confusing symbols, and figures without adequate explanation. Compounding these issues, the absence of a rebuttal leaves all raised concerns unresolved.

**Justification For Why Not Higher Score:**

The experiments are insufficient and not convincing. The size of the dataset is limited. The proposed method performs worse than some compared methods. In addition, it is unclear whether the proposed method is superior to recent methods such as 3D Gaussian Splatting and Voxurf. There are also issues with exposition. All issues raised remain unresolved as there is no rebuttal.

**Justification For Why Not Lower Score:**

N/A

---

### Decision · Program_Chairs · 2024-01-16

Reject